# Knowledge of menstruation and fertility among adults in rural Western Kenya: Gaps and opportunities for support

**Nadia Diamond-Smith**[1]*, **George O. Onyango**[2], **Salome Wawire**[3], **George Ayodo**[2]

**1** University of California San Francisco, San Francisco, California, United States of America, **2** Jaramogi Oginga Odinga University of Science and Technology, Bondo, Kenya, **3** Independent Research Consultant, Nairobi, Kenya

* nadia.diamond-smith@ucsf.edu

## Abstract

An understanding of menstruation and its relationship to fertility can help women know the gestational age of any pregnancies, and thus identify preterm births. It can also help women avoid unintended pregnancies. However, little is known about women, and especially men's, menstruation and fertility knowledge, outside of research on adolescent girls and stigma, and in low and middle income countries (LMIC). Additionally, little is known about practices surrounding the tracking of menstruation and fertility, and how, if at all, women would like to be supported in this. This research is the first phase in adapting a support tool for women in a LMIC, using an implementation science approach to understand relevant cultural needs. We explored women and men's understanding of the relationship between menstruation and fertility, and their interest in support tools, through in-depth qualitative interviews in rural western Kenya. We interviewed 45 adult men, adult women and adolescent women all who had children in 2018. We found high levels of misinformation about menstruation and fertility, with most respondents not knowing the correct times when a woman could become pregnant. Common sources of knowledge included friends/family and school. Few women got information from health providers, even when they were at a facility already for care. There were mixed feelings from women about wanting support from male partners regarding tracking menstruation. While women were interested in a tool that could help them track their menstruation and pregnancies, they had privacy concerns about a mobile health app approach and preferred simpler calendar based tools. This study provides evidence for the high need for correct menstruation information among both men and women, and not only for adolescents. It also suggests that despite the international health community's enthusiasm for mobile health solutions, that approach might not be most appropriate for this topic and setting.

## Introduction

Understanding the relationship between menstruation and fertility is essential for helping women (and their partners) plan pregnancies and avoid unintended pregnancies [1,2]. It also

**Data Availability Statement:** Data are not publicly available due to the qualitative, personal, nature of the data, however, if someone is interested they

can contact Dr. Nicole Santos (nicole.santos@ucsf.edu) who can share anonymized transcripts.

**Funding:** This grant was funded by the Preterm Birth Initiative at the University of California, San Francisco through a larger grant from the Bill and Melinda Gates Foundation. NDS and GO received the grant. The funder did not play a role in the study design, data collection or analysis, decision to publish or preparation of the manuscript.

**Competing interests:** The authors have declared that no competing interests exist.

can empower women with knowledge about their bodies and health. Additionally, knowledge of when in a woman' cycle she becomes pregnant and general awareness and tracking of menstruation allows for women to know their last menstrual period (LMP) and subsequent gestational age when she seeks antenatal care (ANC) or is in labor. This is especially important in settings where most women seek ANC later in pregnancies, when gestational age is harder to estimate, even with more advanced technology such as ultrasounds. In western Kenya, 77% of women have their first ANC visit later than 20 weeks of gestation (5 months of pregnancy) [3]. Knowledge of gestational age is necessary for identifying preterm labor and births. Preterm births are of serious public health concern in much of the world, including in Kenya, where an estimated 193,000 preterm births occur annually [4]. Preterm birth is associated with short-term health risks (including mortality), as well as long-term health, cognitive, social and economic disadvantages [5,6]. This study is nested within a larger multi-country study on preterm births (The Preterm Birth Initiative) and explores knowledge and practices around menstruation and menstruation tracking among parents who experienced a preterm birth. The overall goal was to inform the development of an intervention.

The 2014 Demographic and Health Survey of Kenya found that only 26% of women knew the correct fertile window, showing no change from the previous 2008 Demographic and Health Survey [3]. Other than this, we know little in depth information about women's knowledge of menstruation, the link with fertility, and support and approaches for tracking menstruation. Such knowledge could help with identifying LMP, gestational age, and, ultimately, preterm births. Much of the limited evidence on menstruation knowledge, some of which is from Kenya, has focused on the experiences of adolescent girls, especially related to the impact on school attendance. This literature has focused on adolescent's experiences of stigma or fear about menstruation, and the support and information (or lack thereof) that they received [7,8]. One study in Kenya found that adolescents were not prepared for the start of menstruation, had little information, and did not receive much support from family or school [7]. A recent systematic review highlighted the paucity of data about menstruation among non-adolescent women [9].

More broadly, previous studies in Kenya have found that adolescents, and women in general, did not receive the high quality required information that can enable them make informed decisions related to sexual and reproductive health [10,11]. Some of the barriers to young people receiving reproductive health education lie in provider lack of comfort, both from a provider/counselling, as well as cultural/religious, perspective [12]. Another study in Kenya found that girls desired communication from their mothers about menstruation, but this was generally poor [13]. A study in urban Nairobi found that the most common sources of information about sexuality and contraception were parents/family, followed by school [11].

Not only do women have low levels of knowledge about menstruation, but men do too. Past research in rural Kenya found that two thirds of women received menstrual hygiene support (in the form of menstrual pads or money for buying pads) from their sexual partners, suggesting that male partners also have a role to play in supporting women's sexual and reproductive health practices [14]. Men's own knowledge of fertility and menstruation is also likely important for planning pregnancies, however, little research has explored men's knowledge and perspectives other than research focused on contraception, which has found that men think of reproductive health as a women's domain and generally have low levels of knowledge [15].

We know little about the experiences, knowledge, and practices regarding menstruation among adults, including both men and women, as male partners may act as key sources of financial support or barriers. The aim of this study was to understand women and men's awareness of the link between menstruation and pregnancy, sources of information about

menstruation and fertility, and menstruation tracking practices. We followed an implementation science framework, specifically the Knowledge to Action for Public Health Framework [16], which outlines three main phases: research, translation, and institutionalization. To inform the research phase of this process, we sought input from women and men about how to better support them in menstruation tracking, and what potential tools could be developed that would be most culturally appropriate, acceptable, and desirable. The research presented in this paper is the first phase of this framework, at the discovery level [16]. We were specifically interested in views on tools using mobile technology, given the spread of mobile devices and also fertility/menstruation tracking apps globally. A 2016 review found 108 free menstrual tracking apps, although this was focused on those that were in English and for smartphones [17]. One study in Northern Kenya by Kazi et al (2017) found that between 82–99% of women attending a clinic had access to a mobile phone and about 90% were interested in receiving SMS's about prenatal care or child vaccinations [18]. Other than this, little is known about interest and viability of mobile tools in a rural, LMIC setting such as rural Kenya.

## Methods and materials

This qualitative study was carried out in 2018 among the communities around the shores of Lake Victoria in Bondo, Siaya County, Western Kenya. Bondo was chosen because the Ministry of Health identified this county as having the highest rates of preterm birth, especially rural areas, where there are high levels of fertility, infant mortality, illiteracy and poverty. These results are part of a larger study on preterm birth. Prior to the study, we sought authorization from the Bondo Sub County public health administration, local Ethical Review Committee and had a pre-visit to meet the leaders of community to create a rapport and to familiarize with the study area.

Public health facilities were chosen because most of the population seeks care from these facilities. The study was at three public health facilities in Bondo Sub County, based in rural settings. An initial stakeholder meeting was held with county and sub-county health officials.

Parents were included for participation in the study if they were (1) a female between the ages of 15 and 49 and had at least one child born preterm who was alive and less than 5yrs of age (2) male over the age of 20 and had at least one child born preterm. Health workers helped identify parents of preterm infants. In-depth interviews were conducted with 45 participants: 15 adolescent mothers (age 15–20), 15 adult mothers (21–49) and 15 fathers (over age 20), all with children under 5 years. Male adolescents were excluded on the basis of expert opinion from within the study team; both recruitment rates and adolescent male involvement with preterm infants were anticipated to be low.

Potential respondents were approached by a research coordinator at each facility and read an informed consent and description of the study. If interested, they gave written informed consent. The interviews were conducted by trained research assistants (RAs), in locations convenient for the respondent–including at health facilities and at their homes. Individual interviews lasted an hour on average. The interviews were done in English, Kiswahili and Dholuo, depending on which language was preferred by the respondent. The interviews were conducted in private rooms away from the people in the facilities and anonymity of the participants upheld by not referring to the identities or names of the respondents. The data were recorded through note taking and audio-recording, and then transcribed and translated before being read and cleaned.

Data Analysis: We used a content analysis approach to analyzing our data, with a mixture of both inductive and deductive coding [19]. The transcripts were exported into Atlas Ti qualitative software [20]. A team of four researchers (3 Kenyans and 1 from the US) developed an

initial codebook based on the interview guide. Then one interview was read and coded by all researchers, and additional codes were added. Another set of interviews was coded by the full team, and then the codebook was again edited. A final set of interviews was coded by all researchers, and the final codebook agreed upon. After that, all interviews were coded by only one member of the research team. Data was analyzed in groups of respondent characteristics: adolescent mothers (age 15–20), older mothers (age 21–49) and fathers (over age 20).

The themes were compared across the transcripts and specifically the different groups, to establish the range and similarities of the participants' perceptions, experiences and views. Narrative texts were applied around the themes, with verbatim quotes used to illustrate the text and effectively communicate its meaning.

This study received human subjects approval from the Institutional Review Board at the University of California, San Francisco, USA and Jaramogi Oginga OdingaTeaching and Referral Hospital Ethical Review Board, Kenya. Participants gave written consent.

## Results

### Demographics

The study involved adolescent mothers, mothers aged 20–49 and fathers all who had children below 5 years. Adult women were on average 25, adult men 35 and adolescent women 18 years old (Table 1). Most women had primary education (60%), and none of the adolescent women had tertiary, although this could have been a reflection of their young age or interrupted education due to childbearing. More men completed secondary or tertiary education than women. All respondents were Christian. Adult women had a mean of 2.1 children, adult men 3.3 children and adolescent women 1.3 children. A majority of the inhabitants were involved in fishing, small scale entrepreneurship or subsistence farming as sources of livelihood. A small proportion of the residents were in formal employment such as teaching, nursing, civil service and other services. Poverty levels were high with most of the residents surviving on less than a dollar a day.

### Sources of information about menstruation

Respondents received most of their information through school, non governmental organizations (NGOs), family or friends. Friends were the most common source of information for women, who at times received accurate information from friends, more so than from school. Commonly, when a girl first started getting her menses, other girls told her that this meant she could now become pregnant. As is clear in the quote from the following woman, confusion mixed with underlying shame made it hard to reach out to friends for information and support:

**Table 1. Participant demographics.**

| Parameter | Adult women (n = 15) | Adolescent Women (n = 15) | Adult Men (n = 15) |
|---|---|---|---|
| **Age** (mean, range) | 25, 20–31 | 18, 16–19 | 35, 24–55 |
| **Education** | | | |
| Primary | 60% (n = 9) | 60% (n = 9) | 46.7%(n = 7) |
| Secondary | 26.7% (n = 4) | 40% (n = 6) | 46.7% (n = 7) |
| Tertiary | 13.3% (n = 2) | 0 | 6.7% (n = 1) |
| **Religion** | | | |
| Christian | 100 (n = 10) | 100% (n = 15) | 100% (n = 15) |
| **Number of children** | | | |
| (mean, range) | 2.1, 1–3 | 1.3, 1–3 | 3.3, 1–6 |

I would want to hide it from her (my friend) because I feel embarrassed. It is just embarrassment because the first day I talked to her about it I did not know what was happening to me. I could hear us being taught but I did not think I had reached that age. So I was forced to ask her. She explained to me and told me I had attained that age and so every month I would have my periods and in case I missed my periods any month it would mean I had become pregnant. (Maria, 28, married woman, 2 children)

Women also mentioned talking with other women about menstruation, and its link to fertility, most frequently co-wives or sisters.

I normally ask her (co-wife) because she gets pregnant frequently hence make her have many children. I do ask her why she gets pregnant like almost immediately after delivery. And she tells me that immediately after delivery she gets her monthly period. I talk to her about such kinds of things and I tell her that after giving birth to my baby who is 8 years now, I stayed 4 years without getting my monthly periods (Mona, 25, married woman, 3 children)

Women who felt that they had not received information blamed lack of close female friends or relatives, for example, in the quote below where a woman lived with her grandmother and therefore did not have information until she learned about it in school.

The problem is that I grew up with my grandmother and she never told me anything about menstruation. At times a girl could soil her dress then I would ask to know what happened. But the school contributed a lot in teaching me about menstruation. (Akoth, 23, Married woman, 3 children)

A few men discussed how they learned about menstruation from their wives. A few described how, since they had been married at such a young age, they were already married when their wives first experienced menses and they told their husbands about it, so they learned of it then too. Other men mentioned male friends and elders as key sources of information.

I asked one of my friends if women experience menses and he told me that women must experience menses. Every month a woman must experience menses and a woman who is not experiencing menses could be having a problem, you need to take her for treatment. I then asked him that what problem could be there. He told me that women's problem are so many, you may need to look for old men to come and tell you issues regarding marriage and how marriages are, that is how he left me. (John, 25, married man, 1 child)

One man discussed in detail the information that he had received in school, which was actually misinformation about the relationship between menstruation and pregnancy.

R: In school we were being taught during a subject called health science. We were told that when a woman is on her menses, a few days to end the menstrual cycle if she engages in sexual intercourse with a man then she can get pregnant. . . When she is about to clear her menses.

I: When she is about to clear her menses?

R: I mean if she clears today and tomorrow if she has sex then she can get pregnant

I: You were taught in health science that if a woman finished her menses immediately after that if she engages in sex then she can get pregnant?

R: Yes (Owiti, 55, married man, 5 children)

The adolescent mothers, who were mostly not married at the time they became pregnant, discussed talking to their boyfriends about menstruation as a way of planning when they would "visit" each other (in this population many of the boyfriends seemed to be working elsewhere, so the girls would visit them, and have sex, only at scheduled times).

I: Do you talk to him (boyfriend) about your monthly period?

R: We usually talk, in fact even when this one began and went on for one week, I told him my period had gone on for a whole week in the previous month and this month I had not got my period. He told me to just wait

I: You told me you use a calendar. Does your boyfriend help you track your period?

R: No, he lives far. We meet once in a while. But when I first started having my period he knew the dates; if I told him he would know. After delivery I hear your cycle can change

I: So how would he react when you would tell him about your menses before you delivered?

R: He would tell me when I know my period is close I should not visit him [laughter] because anything can happen

I: What did he mean by that?

R: We can get a baby (Sharon, Adolescent mother, age 18)

Another potential source of information about menstruation, especially the link between fertility and menstruation could have been at health facilities, or even at the time of antenatal care visits (ANC) or delivery. However, in our study while most women were asked about their LMP during their first ANC visit, they received no other related information, even why they were being asked that question. Younger women were more likely to go to their first ANC visit later in pregnancy, and they cited denial about being pregnant, fear of being scolded by staff or being attended to by male staff, and a general belief that people looked down upon them if they went to the health facility too early, as reasons not to go.

The reason why I feared (to go for ANC) when I was expecting my baby, is that there were male staff at the facility. I was imagining that I was going to find the same male staff there, so it was making me more worried. When I was expecting baby in 2007, I just remembered the male staff and it was not easy for me. But when I came I found there were some changes . . .The male staff were not there, there were sisters, and they were sisters that you can even share with them your ideas. (Auma, 21, married woman, 2 children)

I went (to the clinic) later when the pregnancy was 5 months. . .. I was afraid because I was going to a certain clinic in Kibera and they were very harsh. . .The clinic staff, they wanted the exact date when one conceived, which I did not know. And that is what made me fear. That is what made me take long I went after 3 months then stayed much longer before going back. (Awino, 24, married woman, 2 children)

## Knowledge about the relationship between menstruation and pregnancy

Although most respondents knew there was a relationship between menstruation and pregnancy, accuracy of information was very poor. A few respondents did generally have the

correct information, however, they were a little unsure and hesitant about the exact details. As one woman explained:

> R: What I was taught in school or what I hear from my friends is that you can count your days like if your menses end today you can have sex like 3 days without using anything and you don't get pregnant, I don't know 3 or 6 days. There are some days after your menses that you cannot get pregnant, they are called safe days something of the sort. (Achieng, 31, married woman, 1 child)

Most respondents believed women could become pregnant during menstruation or in the few days before and after. Men especially held this belief, and described how they specifically avoided the days during their partner's menstrual cycle, and perhaps the few days after that.

> She always tells me when she is experiencing menstrual cycle. She tells me that "today I am experiencing menses and when I am experiencing the same you cannot have sex with me without protection unless you leave me to complete my days or you use a condom". I also took note that when she is on her menses we don't have sex, I leave her until she completes those her days after four days is when we start having sex with her. (Ojwang, 25, married man, 1 child)

Another man explained that he also believed women could become pregnant the few days before her menstruation, as well as during and after.

> R: To get pregnant for a mother in that menstrual period, with my understanding it is before seeing that blood and after seeing that blood before seven days.
>
> I: Can you come again?
>
> R: It is that before the mother sees that blood, the blood will come and by that time it is still not bleeding and maybe you are having sex and it will find when that egg is mature and in that pregnancy maybe an outcome. Sometimes the woman has experienced periods and has completed the periods, but before seven days after ending the periods, she can also conceive at that time. (Odhiambo, 26, married man, 2 children)

Interestingly, men and women acknowledged some confusion about the relationship between the timing of menstruation and when a woman could get pregnant, discussing how they had heard different information from various sources. As seen in the quote below, some respondents, especially men, did use terms that were stigmatizing about menstruation, such as that a woman was "dirty."

> Some men are the ones who do say that if a woman is experiencing her menses that is the time he needs to have sex with her to make her pregnant and some are saying that when she is on her menses she is dirty, such that he cannot have sex with the woman. (Ojwang, 25, married man, 1 child)

Women also expressed fear and stigma about their menses, especially fear that it would occur while they were at school or other public settings.

> R: I used to hear that people 'dhi e dwe' {Get monthly periods } and used to wonder what that is. When we were going to school some girls could have blood stains at the back of

their dresses and that used to worry me a lot. There was a time when I had gone to visit my Aunt and it happened to me. This made me worry a lot but since that time I knew that it's a stage that a girl must pass through.

I: You have said that whenever a girls dress would be stained at the back it would cause you a lot of worries.

R: Yes.

I: What was making you get so worried?

R: It was worrying because at times it could happen when you are in class and the boys would make fun of it so I used to imagine that the same thing would happen to me. (Mona, 25, married woman, 3 children)

## Tracking of menstruation and support

Many women tried to track their menstruation, primarily calendars. Some men and women both said that they memorized the date. Some women specifically asked their partners to help them remember the date. Additionally, many women struggled with irregular periods and some stated that they did not know the date of their last menstrual period (LMP) when they became pregnant, highlighting the need for support for women in tracking their periods.

While most men were open to talking with their partners about menstruation, some men felt that tracking menstruation was a woman's responsibility, as explained by this man below, who was using "safe days" with his partner.

I: How were you using safe days?

R: Safe days I think being a woman they know how to calculate this, they know the time they are safe.

I: So it is the woman who was calculating not you?

R: It is the woman,

I: So she was telling you this time no, this time it is okay to have sex?

R: Yes (Otieno, 35, married man, 5 children)

Some husbands discussed the importance of knowing about their wives' menstruation so they could support their wives, for example, by buying pads. As the same respondent as the quote above said: "We talk, we discuss it because sometimes I take the responsibility of buying pads and if you cannot be discussing with her, then I don't think you could be at a position to buy pads." (Otieno, 35, married man, 5 children) One man even linked the importance of communicating about menstruation with the ability to plan pregnancies and know the date that the pregnancy occurred.

R: It is important if you are free with your wife talking to her, both of you are free with each other and she can tell you anything on her body and you can also tell her anything on your body, but I think it is important for one to know.

I: Why do you think it is important?

R: Because it is important when you know her timing when she is experiencing her menstrual cycle than when you don't know. When you know time she is experiencing her

menses, then you have sex and she get pregnant and you know that date she has gotten pregnant, you know there are women who are not even aware when they get pregnant and the pregnancy will come as a surprise to her until she is told by other women that are you pregnant and she will say that I am not pregnant. That is when you have a man who is "ma rikni" [proactive] you will hear him telling you that you know you are pregnant. (Ojwang, 25, married man, 1 child)

Aside from husbands who wanted to know about their wives' periods in order to help buy them pads, much of the discussion about husbands wanting to know in order to know if they could have sex. One woman discussed that her husband thought she was using her period as an excuse and made her show him the evidence (blood) before believing her. A husband described needing to know because of superstitions about not having sex with other women during his wife's menstrual period.

R: I should know that she is experiencing menstrual cycle, because according to us Luos it is a way of protecting one's self, so I cannot move out anyhow.

I: How can you not move out anyhow?

R: I cannot meet and have sex with every woman, according to Luos it is a way of protection or "tieko kwer" [Cleansing a taboo]. . .

I: So you cannot have sex with any other woman?

R: Yes I will be waiting for my wife.

I: What happens if you have sex with another woman?

R: I can "roche"[Do something considered a taboo that can affect her reproductive health]

I: In what way?

R: She may then not give birth. (Obwogo, 25, married man, 1 child )

Women were interested in simple tools to track menstruation, and highlighted a need for privacy. Many women were interested in simple tools like a journal or calendar (on paper) that they could mark in, with some noting that illiterate women could potentially use that. Some women already used their phones, by setting an alarm (although other women felt that an alarm was a potential breach of privacy). Some had concerns of a phone because could easily be lost or stolen or that someone else could see phone, including children, who were seen as more technology savvy.

You know our kids nowadays they have become active especially on phones because they would want to know almost everything and they would want to know everything on the phone. So if you put it in a coded way then maybe the kid won't understand it easily, but for the two of you, you will know the meaning of that sign or whatever you will be putting on the phone. (Otieno, 35, married man, 5 children)

Other respondents saw the benefit of a phone since it was something they already had and were using, and that it had the potential to provide other types of health related information as well. A few respondents even gave suggestions about how a phone could be used to help women, as described below.

Yes when you hear that alarm ringing in the phone and you see the picture of a moon indicating, it will act like a message, when you open that message you get all the information that you should be prepared like this, this is what the menstruation period says, you should act like this, you should be clean, pads it tells you like that. (Oguda, 26, married man, 2 children))

Overall, however, due to privacy concerns and lack of comfort, most female respondents felt that something like a calendar that they could mark more privately and more easily hide would be more beneficial to them. Additionally, few respondents had smart phones, and therefore even if there was a mobile option, it would have to be very simple. Notably, a few of the adolescent mothers worried that their parents would be suspicious of something unusual on their mobile phones, and therefore felt that it was not a safe option.

## Discussion

Women and men in rural Western Kenya have low levels of knowledge about the relationship between woman's menstrual cycle and pregnancy, and high levels of misinformation about the timing. Furthermore, the source of information is not often a health provider or school, but rather family or friends. Our findings support other studies from LMICs that women most often seek information from female family members, and that these relatives, as well as other sources such as teachers in schools, are not able to provide accurate information [21]. Better education to both men and women, perhaps through standard approaches of school, but also utilizing times when women are already at health clinics, such as for ANC, could help improve understanding. The current school curriculum taught in Kenya introduces basic human reproduction or sexuality related content in upper primary levels and secondary education detailed reproductive health education is covered. This relates to knowledge on sexuality and high rates of pregnancy among early education drop outs compared to those with secondary and higher education. More detailed information may be needed in both primary and secondary schools on these topics. The first ANC visit, since LMP should already be discussed, may be a missed opportunity to educate women about why providers ask about, and the importance of, gestational age. The time of delivery, family planning or any other postpartum visits (which should occur but rarely do) could also provide opportunities to provide women more detailed information about their fertile window and menstruation more generally. However, much information already needs to be covered at all of these health care visits, potentially limiting these visits as viable information exchange opportunities. Fear of poor person-centered interactions with health care providers, or past negative experiences, appear to be contributing to late ANC attendance, and can impact future health care utilization, again highlighting the need to consider other avenues for education. Community level health workers and facilities and social networks are currently the dominant source of information, so finding mechanisms to improve knowledge among the nodes of social influence is key. Previous studies have called for strengthening community health workers, however, there is little evidence as to the efficacy of this approach [21].

Our findings add to the existing literature on stigma around menstruation, which previously was collected only from adolescent girls—this is clearly an issue that spans age and sex. Underlying stigma and norms from both men and women about menstruation being "dirty" or it being a woman's responsibility are challenges to developing support mechanisms that can involve both men and women. Furthermore, there does appear to be a possibility for men to use knowledge of menstruation as a means of controlling women's fertility, fidelity, or behaviors. Thus, approaches to provide support to women must include both options of privacy as

well as opportunities to engage their partners if desirable. There is a great deal of literature about the need to engage men in reproductive health issues including family planning, HIV/STIs, and ANC and delivery, but we found no discussion of educating and engaging men about menstruation [22–25]. Addressing stigma and gender norms early on with younger generations could help improve the situation for couples in the future.

It is essential to challenge the pervasive wave of excitement about mobile technology to ensure that such approaches are appropriate for women today for the specific health need. Perhaps in the future, as gender norms and stigma around these issues fades, there will be an even greater opportunity to provide support to women through mobile technology. Lower tech options also have great potential to be socially acceptable, easily adapted, and cheaply and widely disseminated and scaled up. Lower tech options should not be neglected as we think about how to best provide information and support to a diversity of women today.

That being said, tools such as mobile phones for helping women (and potentially their partners) track menstruation are of interest to people in this setting. Fears about privacy still persist, and the phone overall feels like a more vulnerable method to most respondents. This highlights the need for personal codes or other carefully designed tools that will ensure privacy. Despite the potential for mobile phones to provide this resource, in this population and for this topic, careful consideration must be given to design.

As with all research, there are limitations to these findings. Data were collected only from one ethnic group in one region of Kenya, and therefore may not be representative of views and experiences of men and women in other communities in other parts Kenya, especially urban areas, or to remote parts of the country. Additionally, all respondents already had children, so they may have had higher knowledge (or lower) than respondents who did not. Also, all respondents had a preterm birth, so might be different, and likely more at risk of having low information about menstruation, than other respondents. Finally, we did not capture the perspectives of male adolescents, who may have more interest in smart phone technology and also differing levels of knowledge than older males or adolescent women. Despite these limitations, this research provides in-depth insight into adult women and men's knowledge and understanding about menstruation and fertility, and how we can better provide information and support.

## Conclusions

Simple tools, both mobile and not, to help women track menstruation, capitalizing on health care contacts and addressing norms and knowledge through social networks have potential for improving knowledge and practices about menstruation and fertility. In our study setting in rural Western Kenya, the smartest option might be very low-tech, and must take into account both technology itself but also comfort with that technology by the population of interest. Empowering women and men in rural setting with accurate information could help reduce unplanned pregnancies and aid in identifying preterm births, as well as providing women with awareness and knowledge about their bodies and health, aside from pregnancy and childbirth.

## Acknowledgments

The authors would like to acknowledge the generous support of the Preterm Birth Initiative at the University of California, San Francisco, funded through the Bill and Melinda Gates Foundation. We would also like to thank Nicole Santos and the rest of the Preterm Birth Initiative team at UCSF for their support. Finally, we would like to thank the participants of the interviews for their time and sharing their experiences.

## Author Contributions

**Conceptualization:** Nadia Diamond-Smith, George O. Onyango, George Ayodo.

**Formal analysis:** Nadia Diamond-Smith, George O. Onyango, Salome Wawire, George Ayodo.

**Funding acquisition:** Nadia Diamond-Smith, George Ayodo.

**Investigation:** George O. Onyango, Salome Wawire.

**Methodology:** Nadia Diamond-Smith, Salome Wawire.

**Project administration:** George O. Onyango, Salome Wawire.

**Supervision:** Nadia Diamond-Smith, Salome Wawire.

**Writing – original draft:** Nadia Diamond-Smith.

**Writing – review & editing:** George O. Onyango, Salome Wawire, George Ayodo.

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
