## [Decision Letter · Decision Letter 0]

24 Oct 2019

PONE-D-19-24517

Knowledge of menstruation and fertility among adults in rural Western Kenya: gaps and opportunities for support

PLOS ONE

Dear Nadia Diamond-Smith,

Thank you for submitting your manuscript to PLOS ONE. After careful consideration, we feel that it has merit but does not fully meet PLOS ONE’s publication criteria as it currently stands. Therefore, we invite you to submit a revised version of the manuscript that addresses the points raised during the review process.

We would appreciate receiving your revised manuscript by November 8th 2019. To enhance the reproducibility of your results, we recommend that if applicable you deposit your laboratory protocols in protocols.io, where a protocol can be assigned its own identifier (DOI) such that it can be cited independently in the future. For instructions see: http://journals.plos.org/plosone/s/submission-guidelines#loc-laboratory-protocols

We look forward to receiving your revised manuscript.

Kind regards,

Violet Naanyu

Academic Editor

PLOS ONE

Journal Requirements:

2. Thank you for stating in your manuscript: "This study received human subjects approval from the University of California, San Francisco, USA and Jaramogi Oginga OdingaTeaching and Referral Hospital, Kenya."

This grant was funded by the Preterm Birth Initiative at the University of California, San Francisco through a larger grant from the Bill and Melinda Gates Foundation. NDS and GO received the grant. The funder did not play a role in the study design, data collection or analysis, decision to publish or preparation of the manuscript 

We note that one or more of the authors are employed by a commercial company: Independent Research Consultant, Nairobi, Kenya

Additional Editor Comments:

This is an important research topic in Kenya. We therefore appreciated this study.

Address each of the comments and queries provided by our reviewers.

Reviewers' comments:

Reviewer's Responses to Questions

**Comments to the Author**

1. Is the manuscript technically sound, and do the data support the conclusions?

Reviewer #1: No

Reviewer #2: Yes

2. Has the statistical analysis been performed appropriately and rigorously? 

Reviewer #1: N/A

Reviewer #2: N/A

3. Have the authors made all data underlying the findings in their manuscript fully available?

Reviewer #1: No

Reviewer #2: No

4. Is the manuscript presented in an intelligible fashion and written in standard English?

Reviewer #1: Yes

Reviewer #2: Yes

5. Review Comments to the Author

Reviewer #1: Abstract

Is concise and neat. Provides a fair sense of what the study was about but the results are brief and provide no hard data about the study population and its characteristics.

Introduction:

Introduction succinct. Discusses well the paucity of research in male and female knowledge of sexual and Reproductive Health. Specifically, it covers well the low level of knowledge of the role of menstruation in SRH. Men's role in managing menstrual periods is highlighted by the fact that they support women financially towards buying of pads etc but they have little knowledge of and interest in of tracking menses. Also highlights common occurrence of mobile phones and interests in using it in RH. Questions if there is similar interest in using mobile phones for tracking menses.

Materials and Methods

Lines 104 thru 106: Indicates that in-depth interviews were conducted with 45 participants 'evenly drawn from ...."what does evenly distributed mean? Provide actual numbers of each type of participant.

Lines 111 -113: Introduces the notion of FGD as a study procedure for the first time. Who were invited for FGDs and who were invited for in-depth interviews? How many participated in the FGDs and how many FGDs?

Overall, the methods section poorly written. Should describe the participants separately for IDI and FGD: re:numbers, enrollment procedures, interview procedures.

Results.

Demographics:

Lines 141 - 147: Is there a table illustrating the numbers referred to in this narrative? The narrative is too general! The findings should start with a description of characteristics of the study participants. Provide.

Sources of Information:

Needs some back up numbers / data

The qualitative data should be a little more synthesized… looks like raw data as currently rendered. Moreover the sources of the verbatim data quoted need to be presented as (XYZ, woman, FGD or IDI)

Discussion

Line 437 -438: “However, despite the potential for mobile phones to provide this resource,

in this population and for this topic, this may not be the most culturally appropriate approach.” The data presented does not suggest any thing cultural but rather a concern about privacy - if phone stolen and fear that children who are techno-savvy could access the information (lines 375 -400)

Line 441 -445 is based on the erroneous interpretation of the data presented. The issue is that of privacy in mobile telephones and the participants even suggest it is possible to go around the challenge through use of codes known to the couple only.

Conclusions

First line is still propagating the falsehood about use of mobile telephones. Rest is OK

No tables or demographics are presented… major weakness of the manuscript. Effectively, we don't know the study population to which this data belongs.

Reviewer #2: 1. General Comments & Recommendations

This is a very interesting and important concept. As a foreigner practicing gynaecology in Western Kenya it is so clear to me that menstrual health illiteracy is high and that this has important consequences for pregnancy care, fertility/infertility, etc.

In the abstract and introduction there are many run-on sentences.

I would love to hear about the set up for this study. Were initial stakeholder meetings required to get permission from the county, sub county, or health facility for this work? Why was this sub-county chosen? These steps are crucially important for building relationships, as I am sure you know, but never get airtime in a manuscript. I think we should be adding them to our methods description, and occasionally our results and discussion.

What was the implementation science framework that you used?

Why were participants chosen if they had a child less than 5 years? Why did they have to be parenting?

I would have liked to see some questions to the adolescent participants around their current relationships with their parents—would they like to receive information about menstruation from their mother? What could that look like? It is an age group that, yes, suffers incredible stigma if they are pregnant; because they are still young and potentially living with their guardian they may still have a potential for intervention as they transition from child to adult.

Did you ask more questions around what kinds of tools the participants would like to track menstruation that are lo-fi/not mobile dependent?

I would suggest that the first ANC visit is not a missed opportunity to teach about the fertile window but maybe an opportunity to teach about why we ask about LMP and how we count gestational age. Because the proportion of women who go to 2 ANC or more is so much lower and because women present so late for prenatal care, all the pregnancy related information is meant to get dumped on them: FP postpartum, delivery in facility, vaccinations, breastfeeding, their blood work and infection screening, the DEET treated mosquito net, malaria prophylaxis, etc etc. I think it is overwhelming for women and for the nurses, and is already inadequate in its delivery. While the 1st ANC is a time when women will come for care and thus serves as a potential intervention time point, I think, practically-speaking, harping on how menstruation works to a woman who can’t remember her period and who is at the end of the 2nd trimester or early 3rd trimester will end up missing its mark and she will forget the information. Maybe it will stick if it can be very succinctly tied to how to think about their fertility after they deliver. Unfortunately, well-woman/preventive health visits are not yet the norm beyond the vaccination program for infants and so I don’t have a clear answer. Women going for FP counselling should certainly receive that teaching. In those clinics, the bleeding irregularities that occur from hormonal contraception are not well explained, which contributes to the myths, misconceptions, and discontinuation rates. In our own group we ran a pilot looking at engaging CHVs to do free urine pregnancy tests with health information and counselling; that could be a means to start this kind of public health education and care.

2. Major Comments

2.1 Are you able to reference the teaching on menstrual health that is provided in schools and at what class level? For example, if the demographic has a low level of completion of education and menstrual health is taught towards the end of primary school, maybe some of them never even had the opportunity to learn in that forum.

2.2 Please comment on inclusion and exclusion criteria for the participants, including age. Were all adolescents age 10 and above considered? Were adolescent boys considered for inclusion? Why or why not? Were only public ministry of health facilities chosen, or were private non-profit or for-profit facilities also included? Why was Bondo sub-county chosen?

2.3 In the Demographics for the results please be a bit more specific if possible, eg. What age range were the adolescent mothers? How many of the adolescent participants were still in school or living with a guardian or married? It is nice to hear who the participants were that you quote or share the ideas from.

2.4 Please describe a bit further the low technology tools the women were interested in

2.5 Please add your conceptual framework/implementation science framework as a Figure

2.6 If there are further themes to share, a schematic could be helpful as a visual summary of your descriptions

3. Minor Comments

3.1 please add references to the first sentence

3.2 consider parsing out the first few opening sentences in the abstract and introduction as they consider more than one idea

6. PLOS authors have the option to publish the peer review history of their article (what does this mean?). If published, this will include your full peer review and any attached files.

Reviewer #1: No

Reviewer #2: Yes: JG Thorne

---

## [Author Response · Author response to Decision Letter 0]

17 Dec 2019

Reviewer #1: Abstract

Is concise and neat. Provides a fair sense of what the study was about but the results are brief and provide no hard data about the study population and its characteristics.

Introduction:

Introduction succinct. Discusses well the paucity of research in male and female knowledge of sexual and Reproductive Health. Specifically, it covers well the low level of knowledge of the role of menstruation in SRH. Men's role in managing menstrual periods is highlighted by the fact that they support women financially towards buying of pads etc but they have little knowledge of and interest in of tracking menses. Also highlights common occurrence of mobile phones and interests in using it in RH. Questions if there is similar interest in using mobile phones for tracking menses.

Materials and Methods

Lines 104 thru 106: Indicates that in-depth interviews were conducted with 45 participants 'evenly drawn from ...."what does evenly distributed mean? Provide actual numbers of each type of participant.

Changed to “…with 45 participants: 15 teenage mothers (under age 20), 15 adult mothers and 15 fathers, all with children under 5 years.”

Lines 111 -113: Introduces the notion of FGD as a study procedure for the first time. Who were invited for FGDs and who were invited for in-depth interviews? How many participated in the FGDs and how many FGDs?

Apologies, initially we also presented data from FGDs, but decided to move that to another paper, so we have removed any reference to FGDs in this paper. 

Overall, the methods section poorly written. Should describe the participants separately for IDI and FGD: re:numbers, enrollment procedures, interview procedures.

Results.

Demographics:

Lines 141 - 147: Is there a table illustrating the numbers referred to in this narrative? The narrative is too general! The findings should start with a description of characteristics of the study participants. Provide.

We have added a table and expanded this section to read “The study involved adolescent mothers, mothers aged 20-49 and fathers all who had children below 5 years. Adult women were on average 25, adult men 35 and adolescent women 18 years old (Table 1). Most women had primary education (60%), and none of the adolescent women had tertiary, although this could have been a reflection of their young age or interrupted education due to childbearing. More men completed secondary or tertiary education than women. All respondents were Christian. Adult women had a mean of 2.1 children, adult men 3.3 children and adolescent women 1.3 children. A majority of the inhabitants were involved in fishing, small scale entrepreneurship or subsistence farming as sources of livelihood. A small proportion of the residents were in formal employment such as teaching, nursing, civil service and other services. Poverty levels were high with most of the residents surviving on less than a dollar a day.” 

Sources of Information:

Needs some back up numbers / data

The qualitative data should be a little more synthesized… looks like raw data as currently rendered. Moreover the sources of the verbatim data quoted need to be presented as (XYZ, woman, FGD or IDI)

Since we have clarified that these were all IDIs, we have not added that information, but have added psedonyms. 

Discussion

Line 437 -438: “However, despite the potential for mobile phones to provide this resource,

in this population and for this topic, this may not be the most culturally appropriate approach.” The data presented does not suggest any thing cultural but rather a concern about privacy - if phone stolen and fear that children who are techno-savvy could access the information (lines 375 -400)

We have changed this to read “However, despite the potential for mobile phones to provide this resource,

in this population and for this topic, careful consideration must be given to design.”

Line 441 -445 is based on the erroneous interpretation of the data presented. The issue is that of privacy in mobile telephones and the participants even suggest it is possible to go around the challenge through use of codes known to the couple only.

We have edited this section to tone down the negativity about mobile options and add in privacy codes as you suggest. 

Conclusions

First line is still propagating the falsehood about use of mobile telephones. 

We have clarified that we mean simple mobile tools or other types of tools. 

Rest is OK

No tables or demographics are presented… major weakness of the manuscript. Effectively, we don't know the study population to which this data belongs.

We have added a table and description of demographics

Reviewer #2: 1. General Comments & Recommendations

This is a very interesting and important concept. As a foreigner practicing gynaecology in Western Kenya it is so clear to me that menstrual health illiteracy is high and that this has important consequences for pregnancy care, fertility/infertility, etc.

In the abstract and introduction there are many run-on sentences.

We have read through and made edits. 

I would love to hear about the set up for this study. Were initial stakeholder meetings required to get permission from the county, sub county, or health facility for this work? Why was this sub-county chosen? These steps are crucially important for building relationships, as I am sure you know, but never get airtime in a manuscript. I think we should be adding them to our methods description, and occasionally our results and discussion.

We have fleshed out the methods to read: “This qualitative study was carried out in 2018 among the communities around the shores of Lake Victoria in Bondo, Siaya County, Western Kenya. Bondo was chosen because the Ministry of Health identified this county as having the highest rates of preterm birth, especially rural areas, where there are high levels of fertility, infant mortality, illiteracy and poverty. These results are part of a larger study on preterm birth. Prior to the study, we sought authorization from the Bondo Sub County Hospital administration, local Ethical Review Committee and had a pre-visit to meet the leaders of community to create a rapport and to familiarize with the study area.

Public health facilities were chosen because most of the population seeks care from these facilities. The study was at three public health facilities in Bondo Sub County, based in rural settings. An initial stakeholder meeting was held with county and sub-county health officials. In-depth interviews were conducted with 45 participants: 15 teenage mothers (age 15-20), 15 adult mothers and 15 fathers, all with children under 5 years. Male adolescents were not included because the study team thought it would be difficult to recruit males under 20 who had a child under 5 and the study team felt they less likely to be involved with care of preterm infants.“ 

What was the implementation science framework that you used? 

We used the Knowledge to Action for Public Health Framework, which we have now cited and discussed. This phase is the first (research) phase. 

Why were participants chosen if they had a child less than 5 years? Why did they have to be parenting?

This was part of a larger study about perceptions and experiences of preterm birth, and thus we wanted to talk to people who had had a birth already. 

I would have liked to see some questions to the adolescent participants around their current relationships with their parents—would they like to receive information about menstruation from their mother? What could that look like? It is an age group that, yes, suffers incredible stigma if they are pregnant; because they are still young and potentially living with their guardian they may still have a potential for intervention as they transition from child to adult.

This would have been a great thing to probe about, and in some of the adolescent interviews they did discuss relationships with parents and grandparents and other family members who supported them in their pregnancies. Unfortunately, we only have a few tidbits about this, and also, we wanted to focus on older adults since they have been so understudied about this topic. 

Did you ask more questions around what kinds of tools the participants would like to track menstruation that are lo-fi/not mobile dependent?

We did not probe specifically about more low-tech options, since we were mostly focused on whether they were doing anything, which is where the discussion of more low tech things like paper calendars were mentioned, and then we probed more about their interested in a mobile option since we were initially thinking this would be a good, scalable, low-cost option. 

I would suggest that the first ANC visit is not a missed opportunity to teach about the fertile window but maybe an opportunity to teach about why we ask about LMP and how we count gestational age. Because the proportion of women who go to 2 ANC or more is so much lower and because women present so late for prenatal care, all the pregnancy related information is meant to get dumped on them: FP postpartum, delivery in facility, vaccinations, breastfeeding, their blood work and infection screening, the DEET treated mosquito net, malaria prophylaxis, etc etc. I think it is overwhelming for women and for the nurses, and is already inadequate in its delivery. While the 1st ANC is a time when women will come for care and thus serves as a potential intervention time point, I think, practically-speaking, harping on how menstruation works to a woman who can’t remember her period and who is at the end of the 2nd trimester or early 3rd trimester will end up missing its mark and she will forget the information. Maybe it will stick if it can be very succinctly tied to how to think about their fertility after they deliver. Unfortunately, well-woman/preventive health visits are not yet the norm beyond the vaccination program for infants and so I don’t have a clear answer. Women going for FP counselling should certainly receive that teaching. In those clinics, the bleeding irregularities that occur from hormonal contraception are not well explained, which contributes to the myths, misconceptions, and discontinuation rates. In our own group we ran a pilot looking at engaging CHVs to do free urine pregnancy tests with health information and counselling; that could be a means to start this kind of public health education and care.

Thank you for this thoughtful point, you are right, there is probably a lot going on in that visit and this is an unreasonable ask. Postpartum care and information is something that we are really interested in trying to improve as well, as it is so neglected globally! We have edited that section to read “The first ANC visit is a missed opportunity to educate women about why providers ask about, and the importance of, gestational age. The time of delivery, family planning or any other postpartum visits (which should occur but rarely do) could be points in time to provide women more detailed information about their fertile window and menstruation more generally.”

2. Major Comments

2.1 Are you able to reference the teaching on menstrual health that is provided in schools and at what class level? For example, if the demographic has a low level of completion of education and menstrual health is taught towards the end of primary school, maybe some of them never even had the opportunity to learn in that forum.

We have added the following to the discussion “The current school curriculum taught in Kenya introduces basic human reproduction or sexuality related content in upper primary levels and secondary education detailed reproductive health education is covered. This relates to knowledge on sexuality and high rates of pregnancy among early education drop outs compared to those with secondary and higher education. More detailed information may be needed in both primary and secondary schools on these topics.”

2.2 Please comment on inclusion and exclusion criteria for the participants, including age. Were all adolescents age 10 and above considered? Were adolescent boys considered for inclusion? Why or why not? Were only public ministry of health facilities chosen, or were private non-profit or for-profit facilities also included? Why was Bondo sub-county chosen?

We have fleshed out the methods to read: “This qualitative study was carried out in 2018 among the communities around the shores of Lake Victoria in Bondo, Siaya County, Western Kenya. Bondo was chosen because the Ministry of Health identified this county as having the highest rates of preterm birth, especially rural areas, where there are high levels of fertility, infant mortality, illiteracy and poverty. These results are part of a larger study on preterm birth. Prior to the study, we sought authorization from the Bondo Sub County Hospital administration, local Ethical Review Committee and had a pre-visit to meet the leaders of community to create a rapport and to familiarize with the study area. Public health facilities were chosen because most of the population seeks care from these facilities. The study was at three public health facilities in Bondo Sub County, based in rural settings. An initial stakeholder meeting was held with county and sub-county health officials. In-depth interviews were conducted with 45 participants: 15 teenage mothers (age 15-20), 15 adult mothers and 15 fathers, all with children under 5 years. Male adolescents were not included because the study team thought it would be difficult to recruit males under 20 who had a child under 5 and the study team felt they less likely to be involved with care of preterm infants.“ 

2.3 In the Demographics for the results please be a bit more specific if possible, eg. What age range were the adolescent mothers? How many of the adolescent participants were still in school or living with a guardian or married? It is nice to hear who the participants were that you quote or share the ideas from.

We have added a table and expanded this section to read “The study involved adolescent mothers, mothers aged 20-49 and fathers all who had children below 5 years. Adult women were on average 25, adult men 35 and adolescent women 18 years old (Table 1). Most women had primary education (60%), and none of the adolescent women had tertiary, although this could have been a reflection of their young age or interrupted education due to childbearing. More men completed secondary or tertiary education than women. All respondents were Christian. Adult women had a mean of 2.1 children, adult men 3.3 children and adolescent women 1.3 children. A majority of the inhabitants were involved in fishing, small scale entrepreneurship or subsistence farming as sources of livelihood. A small proportion of the residents were in formal employment such as teaching, nursing, civil service and other services. Poverty levels were high with most of the residents surviving on less than a dollar a day.” 

2.4 Please describe a bit further the low technology tools the women were interested in

We have added the following “Many women were interested in simple tools like a journal or calendar (on paper) that they could mark in, with some noting that illiterate women could potentially use that. Some women already used their phones, by setting an alarm (although other women felt that an alarm was a potential breach of privacy).”

2.5 Please add your conceptual framework/implementation science framework as a Figure. 

If the editors agree, we can add the Framework now cited.

2.6 If there are further themes to share, a schematic could be helpful as a visual summary of your descriptions

These are the themes we are planning to focus on for this paper. 

3. Minor Comments

3.1 please add references to the first sentence

Done

3.2 consider parsing out the first few opening sentences in the abstract and introduction as they consider more than one idea

Done!

2. Thank you for stating in your manuscript: "This study received human subjects approval from the University of California, San Francisco, USA and Jaramogi Oginga OdingaTeaching and Referral Hospital, Kenya."

We have added “This study received human subjects approval from the Institutional Review Board at the University of California, San Francisco, USA and Jaramogi Oginga OdingaTeaching and Referral Hospital Ethical Review Board, Kenya. Participants gave written consent.”

We have added “This study received human subjects approval from the Institutional Review Board at the University of California, San Francisco, USA and Jaramogi Oginga OdingaTeaching and Referral Hospital Ethical Review Board, Kenya. Participants gave written consent.”

This grant was funded by the Preterm Birth Initiative at the University of California, San Francisco through a larger grant from the Bill and Melinda Gates Foundation. NDS and GO received the grant. The funder did not play a role in the study design, data collection or analysis, decision to publish or preparation of the manuscript

We note that one or more of the authors are employed by a commercial company: Independent Research Consultant, Nairobi, Kenya

We are sorry for the confusion, this is not a commercial company, but rather an individual researcher who was paid as a consultant. 

---

## [Decision Letter · Decision Letter 1]

3 Feb 2020

PONE-D-19-24517R1

Knowledge of menstruation and fertility among adults in rural Western Kenya: gaps and opportunities for support

PLOS ONE

Dear Nadia Diamond-Smith,

Thank you for submitting your manuscript to PLOS ONE. After careful consideration, we feel that it has merit but does not fully meet PLOS ONE’s publication criteria as it currently stands. Therefore, we invite you to submit a revised version of the manuscript that addresses the points raised during the review process.

As you work on this revision, ensure you address all queries fully. Remember to add the missing study details. Pay special attention to suggestions made regarding the methodology section, interpretation of data and discussion of your findings. 

We would appreciate receiving your revised manuscript by February 14th 2020. To enhance the reproducibility of your results, we recommend that if applicable you deposit your laboratory protocols in protocols.io, where a protocol can be assigned its own identifier (DOI) such that it can be cited independently in the future. For instructions see: http://journals.plos.org/plosone/s/submission-guidelines#loc-laboratory-protocols

We look forward to receiving your revised manuscript.

Kind regards,

Violet Naanyu

Academic Editor

PLOS ONE

Additional Editor Comments (if provided):

Thank you for the revised manuscript - it is much better!

Kindly address the remaining minor queries and edits and resubmit.

Reviewers' comments:

Reviewer's Responses to Questions

**Comments to the Author**

1. If the authors have adequately addressed your comments raised in a previous round of review and you feel that this manuscript is now acceptable for publication, you may indicate that here to bypass the “Comments to the Author” section, enter your conflict of interest statement in the “Confidential to Editor” section, and submit your "Accept" recommendation.

Reviewer #1: All comments have been addressed

Reviewer #2: (No Response)

2. Is the manuscript technically sound, and do the data support the conclusions?

Reviewer #1: Yes

Reviewer #2: Partly

3. Has the statistical analysis been performed appropriately and rigorously? 

Reviewer #1: N/A

Reviewer #2: N/A

4. Have the authors made all data underlying the findings in their manuscript fully available?

Reviewer #1: Yes

Reviewer #2: No

5. Is the manuscript presented in an intelligible fashion and written in standard English?

Reviewer #1: Yes

Reviewer #2: Yes

6. Review Comments to the Author

Reviewer #1: Even though the authors have substantially revised the document and addressed my comments, the results are still too heavy on the verbatim quotes. Otherwise I am satisfied.

Reviewer #2: 1. General Comments.

Overall this reads much better than the first iteration. Nice work! I still think the story needs tightening to make it flow between the sections.

2. Major Comments

2. 1 In your methods, please clarify inclusion and exclusion criteria in a more succinct manner. E.g Parents were included for participation in the study if they were (1) a female between the ages of 15 and 49 and had at least one child born preterm who was alive and less than 5yrs of age (2) male between the ages of 20 and ??? and had at least one child born preterm... Prematurity was identified by >>>

2.2 List the age grouping in your data analysis: Participant data was analyzed in 3 separate age categories: (list and defend)

2.3 I think it is important to mention this study was a sub-study of the preterm birth initiative in the background of the manuscript, and to site the relevant study inclusion criteria for the bigger study that affected your subset for this study. It explains why you are only asking parents who had preterm births. Otherwise it is not a clear inclusion criteria even with the background provided. Sure, better menstrual knowledge might help identify preterm births, but why not ask all women and men about menses and other SRH?

2.4 Outline the steps of the framework in the methodology

2.5 Do you have data on how many of your participants had simple or smart phones? Since you are asking them about it that would be a helpful data point. It should be >90% have some sort of phone, but there may be discrepancies e.g by age, or couples share a phone (so who does it really belong to...?)

2.5 I study healthcare provider empathy for pregnancy care. I want to first say that (in lines 250-257) it is not clear to me that women identify the first ANC as a time to learn about menstruation. I still think that is a discussion point from the authors. I don't know your question guide but if it was asked about getting this information at the 1st ANC, you could a) say "when asked about receiving this health information at ANC, women said/felt...". I also want to suggest that you can go ahead and say that a) younger women were more likely to attend their first ANC late, and b) women had negative experiences in ANC or preconceived notions about ANC that affected their willingness to attend. Then you discussion is includes that until ANC is seen as a welcoming environment it remains a challenge to harness it as a time for extra healthcare education.

2.6 Have you thought about having a theme specifically on stigma and myths? even your quote about technology includes this need to keep information about periods hidden from their children.

2.7 I don't know if there is an underlying agenda to develop a smart phone or simple phone tool for menses. I suspect there is given the funder. But I think you can be more bold in the discussion and suggest the smartest tool might actually be very low tech, or it may differ by age/generation. There is no question that the high tech solutions showing promise in urban poor areas are not yet relevant to rural poor areas. They will converge at some point.

2.8 I think it is worth saying that you do not have the perspective of adolescent males in understanding menstrual health as a limitation

2.9 Line 491-493 is run on and does not reflect your paper. The data to me is highlighting misinformation, mixed sources of information, and myths and stigma. It highlights that there is a lot of apprehension about a high tech/phone based tracking tool. And there is nothing about preterm birth although there was a long paragraph about it in the background. The story just needs to be tightened a bit. If you flip the first sentence to be about misinformation, stigma, and a desire about improved information, and then add a second sentence that says something about them being open to tracking tools but low-fidelity tools need to be considered and innovated (why can't an innovation be paper based?), and then jump into your third sentence, I think it would read better.

3. Minor Comments

3.1 Line 48-49: delete the comment about ultrasounds that is in brackets. It is not necessary.

3.2 Line 79: should be a new paragraph

3.3 Line 100: "One study in a different part of Kenya" doesn't read well. Consider referencing its geographic location, eg. "in a study from central Kenya by __ et al, 82-99% of women attending clinic..."

3.4 Line 121-123: this is a run-on sentence and I think it is missing words. It can be something like "male adolescents were excluded on the basis of expert opinion from within the study team; both recruitment rates and adolescent male involvement with preterm infants were anticipated to be low.

Regardless, this is a demographic missed, and other data suggests adolescent males are an important group to include in SRH education, including menstrual health and hygiene, for all the reasons argued in the background and more.

3.5 Line 158: Include the age range for adolescent mothers as you did for adult mothers. Be consistent with the age ranges between the Methods and the Discussion in terms of listing them (you list adolescent age range at Line 119 but not the adults, and the reverse at Line 158). Please also state the age range of the adult fathers.

3.6 Make sure all acronyms/short forms are first spelled out. I am pretty sure line 172 is the first time NGO is mentioned.

7. PLOS authors have the option to publish the peer review history of their article (what does this mean?). If published, this will include your full peer review and any attached files.

Reviewer #1: No

Reviewer #2: No

---

## [Author Response · Author response to Decision Letter 1]

12 Feb 2020

February 12, 2020

To the reviewers,

Thank you for your additional feedback on this manuscript. We have addressed all of your comments below. We are grateful for your time and thoughtful insight into strengthening this paper!

Please let us know if you have additional thoughts or questions, 

Nadia Diamond-Smith

Reviewer #1: Even though the authors have substantially revised the document and addressed my comments, the results are still too heavy on the verbatim quotes. Otherwise I am satisfied.

We have cut a few pieces of the quotes, thank you!

Reviewer #2: 1. General Comments.

Overall this reads much better than the first iteration. Nice work! I still think the story needs tightening to make it flow between the sections.

2. Major Comments

2. 1 In your methods, please clarify inclusion and exclusion criteria in a more succinct manner. E.g Parents were included for participation in the study if they were (1) a female between the ages of 15 and 49 and had at least one child born preterm who was alive and less than 5yrs of age (2) male between the ages of 20 and ??? and had at least one child born preterm... Prematurity was identified by >>>

We have changed this to read “Parents were included for participation in the study if they were (1) a female between the ages of 15 and 49 and had at least one child born preterm who was alive and less than 5yrs of age (2) male over the age of 20 and had at least one child born preterm. Health workers helped identify parents of preterm infants.” 

2.2 List the age grouping in your data analysis: Participant data was analyzed in 3 separate age categories: (list and defend)

We have changed this to read “Data was analyzed in groups of respondent characteristics: adolescent mothers (age 15-20), older mothers (age 21-45) and fathers (over age 20).”

2.3 I think it is important to mention this study was a sub-study of the preterm birth initiative in the background of the manuscript, and to site the relevant study inclusion criteria for the bigger study that affected your subset for this study. It explains why you are only asking parents who had preterm births. Otherwise it is not a clear inclusion criteria even with the background provided. Sure, better menstrual knowledge might help identify preterm births, but why not ask all women and men about menses and other SRH?

We have added the following in the first paragraph of the introduction, but let us know if you think it should come later on “This study is nested within a larger multi-country study on preterm births (The Preterm Birth Initiative) and explores knowledge and practices around menstruation and menstruation tracking among parents who experienced a preterm birth. The overall goal was to inform the development of an intervention.”

2.4 Outline the steps of the framework in the methodology

We have added “We followed an implementation science framework, specifically the Knowledge to Action for Public Health Framework (16), which outlines three main phases: research, translation, and institutionalization. To inform the research phase of this process, we sought”

2.5 Do you have data on how many of your participants had simple or smart phones? Since you are asking them about it that would be a helpful data point. It should be >90% have some sort of phone, but there may be discrepancies e.g by age, or couples share a phone (so who does it really belong to...?)

Unfortunately we do not know this, I wish we had asked!

2.5 I study healthcare provider empathy for pregnancy care. I want to first say that (in lines 250-257) it is not clear to me that women identify the first ANC as a time to learn about menstruation. I still think that is a discussion point from the authors. I don't know your question guide but if it was asked about getting this information at the 1st ANC, you could a) say "when asked about receiving this health information at ANC, women said/felt...". I also want to suggest that you can go ahead and say that a) younger women were more likely to attend their first ANC late, and b) women had negative experiences in ANC or preconceived notions about ANC that affected their willingness to attend. Then you discussion is includes that until ANC is seen as a welcoming environment it remains a challenge to harness it as a time for extra healthcare education.

We have added to this section to read “The first ANC visit, since LMP should already be discussed, may be a missed opportunity to educate women about why providers ask about, and the importance of, gestational age. The time of delivery, family planning or any other postpartum visits (which should occur but rarely do) could also provide opportunities to provide women more detailed information about their fertile window and menstruation more generally. However, much information already needs to be covered at all of these health care visits, potentially limiting these visits as viable information exchange opportunities. Fear of poor person-centered interactions with health care providers, or past negative experiences, appear to be contributing to late ANC attendance, and can impact future health care utilization, again highlighting the need to consider other avenues for education.”

2.6 Have you thought about having a theme specifically on stigma and myths? even your quote about technology includes this need to keep information about periods hidden from their children.

We did consider this option, however, felt that the stigma and myths were intertwined with each of the sections that we did end up not separating them out specifically. We also felt that the previous literature, while mostly focused on adolescents, also highlighted more about stigma and myths, and we wanted to focus on some of the more novel findings/areas of exploration.

2.7 I don't know if there is an underlying agenda to develop a smart phone or simple phone tool for menses. I suspect there is given the funder. But I think you can be more bold in the discussion and suggest the smartest tool might actually be very low tech, or it may differ by age/generation. There is no question that the high tech solutions showing promise in urban poor areas are not yet relevant to rural poor areas. They will converge at some point.

There wasn't an agenda for smart phone specifically, but it was of interest. We have tried to make our statements stronger in the main paragraph on this topic, partially by rearranging and spacing out points. We also added a sentence to the conclusion to make this point again. I hope its OK that we loved your wording of “the smartest tool” and used that “In our study setting in rural Western Kenya, the smartest option might be very low-tech, and must take into account both technology itself but also comfort with that technology by the population of interest.”

2.8 I think it is worth saying that you do not have the perspective of adolescent males in understanding menstrual health as a limitation

We have added “Finally, we did not capture the perspectives of male adolescents, who may have more interest in smart phone technology and also differing levels of knowledge than older males or adolescent women.”

2.9 Line 491-493 is run on and does not reflect your paper. The data to me is highlighting misinformation, mixed sources of information, and myths and stigma. It highlights that there is a lot of apprehension about a high tech/phone based tracking tool. And there is nothing about preterm birth although there was a long paragraph about it in the background. The story just needs to be tightened a bit. If you flip the first sentence to be about misinformation, stigma, and a desire about improved information, and then add a second sentence that says something about them being open to tracking tools but low-fidelity tools need to be considered and innovated (why can't an innovation be paper based?), and then jump into your third sentence, I think it would read better.

We have flipped this as you suggest and cut the first sentence. We have stressed the low-tech options in other sections (discussed above). 

3. Minor Comments

3.1 Line 48-49: delete the comment about ultrasounds that is in brackets. It is not necessary.

Removed

3.2 Line 79: should be a new paragraph

Done

3.3 Line 100: "One study in a different part of Kenya" doesn't read well. Consider referencing its geographic location, eg. "in a study from central Kenya by __ et al, 82-99% of women attending clinic..."

Changed to “One study in Northern Kenya by Kazi et al (2017) found that between 82-99% of women attending a clinic had access to a mobile phone and about 90% were interested in receiving SMS’s about prenantal care or child vaccinations (18).”

3.4 Line 121-123: this is a run-on sentence and I think it is missing words. It can be something like "male adolescents were excluded on the basis of expert opinion from within the study team; both recruitment rates and adolescent male involvement with preterm infants were anticipated to be low. Regardless, this is a demographic missed, and other data suggests adolescent males are an important group to include in SRH education, including menstrual health and hygiene, for all the reasons argued in the background and more.

We have changed this and noted this limitation

3.5 Line 158: Include the age range for adolescent mothers as you did for adult mothers. Be consistent with the age ranges between the Methods and the Discussion in terms of listing them (you list adolescent age range at Line 119 but not the adults, and the reverse at Line 158). Please also state the age range of the adult fathers.

Done

3.6 Make sure all acronyms/short forms are first spelled out. I am pretty sure line 172 is the first time NGO is mentioned.

Done

---

## [Editor Report · Decision Letter 2]

18 Feb 2020

Knowledge of menstruation and fertility among adults in rural Western Kenya: gaps and opportunities for support

PONE-D-19-24517R2

Dear Dr. Nadia Diamond-Smith,

We are pleased to inform you that your manuscript has been judged scientifically suitable for publication and will be formally accepted for publication once it complies with all outstanding technical requirements.

With kind regards,

Violet Naanyu

Academic Editor

PLOS ONE

---

## [Editor Report · Acceptance letter]

20 Feb 2020

PONE-D-19-24517R2 

Knowledge of menstruation and fertility among adults in rural Western Kenya: gaps and opportunities for support 

Dear Dr. Diamond-Smith:

I am pleased to inform you that your manuscript has been deemed suitable for publication in PLOS ONE. Congratulations! Your manuscript is now with our production department. 

With kind regards,

on behalf of

Prof. Violet Naanyu 

Academic Editor

PLOS ONE